# Perioperative Acetaminophen Dosing in Obese Children

**DOI:** 10.3390/children10040625

**Published:** 2023-03-27

**Authors:** Brian Joseph Anderson, Luis Ignacio Cortinez

**Affiliations:** 1Department of Anaesthesiology, University of Auckland, Park Road, Auckland 1023, New Zealand; 2División Anestesiología, Escuela de Medicina, Pontificia Universidad Católica de Chile, Santiago de Chile 8331150, Chile

**Keywords:** pharmacokinetics, pharmacodynamics, acetaminophen, paracetamol, pediatrics, drug dosing, allometry, obesity, anaesthesia

## Abstract

Acetaminophen is a commonly used perioperative analgesic drug in children. The use of a preoperative loading dose achieves a target concentration of 10 mg/L associated with a target analgesic effect that is 2.6 pain units (visual analogue scale 1–10). Postoperative maintenance dosing is used to keep this effect at a steady-state concentration. The loading dose in children is commonly prescribed per kilogram. That dose is consistent with the linear relationship between the volume of distribution and total body weight. Total body weight is made up of both fat and fat-free mass. The fat mass has little influence on the volume of distribution of acetaminophen but fat mass should be considered for maintenance dosing that is determined by clearance. The relationship between the pharmacokinetic parameter, clearance, and size is not linear. A number of size metrics (e.g., fat-free and normal fat mass, ideal body weight and lean body weight) have been proposed to scale clearance and all consequent dosing schedules recognize curvilinear relationships between clearance and size. This relationship can be described using allometric theory. Fat mass also has an indirect influence on clearance that is independent of its effects due to increased body mass. Normal fat mass, used in conjunction with allometry, has proven a useful size metric for acetaminophen; it is calculated using fat-free mass and a fraction (Ffat) of the additional mass contributing to total body weight. However, the Ffat for acetaminophen is large (Ffat = 0.82), pharmacokinetic and pharmacodynamic parameter variability high, and the concentration–response slope gentle at the target concentration. Consequently, total body weight with allometry is acceptable for the calculation of maintenance dose. The dose of acetaminophen is tempered by concerns about adverse effects, notably hepatotoxicity associated with use after 2–3 days at doses greater than 90 mg/kg/day.

## 1. Introduction

The dose of any drug is determined through an understanding of pharmacokinetics (what the body does to a drug) and pharmacodynamics (what the drug does to the body). Acetaminophen pharmacodynamic effects include both beneficial (analgesia and temperature modulation) and adverse (hepatotoxicity). These two pharmacodynamic effects modulate the dose. The dose prescribed should be that determined to achieve a plasma concentration associated with beneficial effects, but without untoward effects. Both of these pharmacodynamic effects are concentration-related, rather than dose-related.

Acetaminophen dosing in the perioperative period is complicated by two further aspects. Children are often administered a loading dose before a surgical procedure to achieve that “sweet spot” concentration where beneficial effects outweigh toxic effects, known as the target concentration. That dose, invariably determined by volume of distribution, differs from the maintenance dose that maintains the target concentration in plasma. The other aspect complicating acetaminophen dosing is that there is a fear that doses above a prescribed amount (e.g., 1000 mg four times daily in an adult) contributes to toxicity. This restriction on dose means that the target concentration associated with analgesic benefit can seldom be achieved in an obese teenager [1].

## 2. Current Acetaminophen Dose Estimation

Expert opinion presumes that the dose in the obese child is best determined by pharmacokinetic understanding [2,3,4]. It is known that obesity has an effect on acetaminophen volume of distribution, notably through dissemination between lean body mass and fat mass, both of which increase independently in the obese individual. Drug lipophilicity is thought to be a primary determinant of the volume of distribution to the fat component of body composition [5]. Clearance is assumed to increase, reflective of increases in lean body mass, which in turn is consistent with metabolic activity [6,7,8]. Altered hepatic protein production and consequently altered protein binding in obese children can be an issue for some drugs, but acetaminophen has low protein binding and this is not a major concern for acetaminophen pharmacokinetic changes. Hepatic disease is, however, an issue. Obese adolescents have an increased prevalence of non-alcoholic fatty liver disease that may increase hepatic cytochrome P450 (CYP) 2E1 expression, an enzyme responsible for the production of metabolites that contribute to acetaminophen hepatotoxicity [9]. Morbidly obese individuals may exhibit increased CYP2E1-mediated oxidation of acetaminophen [10], although the clinical impact of this observation had been questioned [11].

Hepatic dysfunction, a consequence of morbid obesity, could have a major effect on acetaminophen clearance. However, clinical scoring systems (e.g., Child–Pugh or MELD classification) are hard to quantify and relate to altered drug disposition. Each patient has an individual pattern of dysfunction and different drugs are affected differently, depending on their clearance mechanism [12]. Altered hepatic blood flow in severe liver disease may, for example, reduce the clearance of drugs that are perfusion-limited, but this pathology has a limited impact on drugs such as acetaminophen that are capacity-limited.

Few of these obesity-related pharmacokinetic changes have been quantified in either adults or children [13]. Acetaminophen pharmacokinetics in overweight or obese children are ill-described and there is little information to guide physicians with regard to the dose [14]. There are some data to suggest that pharmacokinetic parameters are similar in both lean and obese adolescents [15] or adults [16] after single-dose therapy. The obese adolescent has been the subject of commentary [13], but dosing recommendations for the obese adolescent remain uncertain, with suggestions for the use of ideal body weight [17] or lean body weight [18]. These recommendations do not consider the impact of fat mass, do not define at what degree of obesity they should be implemented, do not distinguish between loading dose and maintenance dose, and have never been validated for acetaminophen in children. They do, however, reduce the dose, when expressed per kilogram, as weight increases.

## 3. Size Model Foibles

Total body weight is commonly used for dosing in obese children, but that contributes to dose error because the contribution from the fat mass portion of body composition is seldom acknowledged. The fat mass may contribute to the volume of distribution (V) through drug lipid solubility altering disposition. It is also a metabolically active component of body mass and as such contributes to clearance (CL). Failure to account for fat mass contributes to dosing errors. There are few practical dose recommendations for obese children [2,19] even though it is known that fat mass influences the volume of distribution and clearance [20], that effects from fat mass are drug-specific [20], and that dosing per kilogram (weight-based linear dosing) is a contributor to dose inaccuracies. Further, obesity can be considered an inflammatory disease that both contributes to disease processes and is sometimes consequent to disease processes. An assortment of body size scalers (e.g., total body weight, body surface area, ideal body weight, lean body mass, adjusted body weight, body mass index, fat-free mass, and allometry) have been used to determine the dose in obese individuals [21]. This assortment leaves clinicians confused about which metric to use for a particular drug for an individual child, or even the degree of obesity that commands a size metric other than total body weight. The dosing of children in the operating room may be dependent on a size metric that differs with anaesthesia phases (e.g., lean body mass for propofol induction dose and total body weight for maintenance dose rate) [22,23,24].

Although recommendations for particular size scalers abound in the literature [21], most carry the caveat that the dose in the obese child will be determined by better pharmacokinetic understanding [2,3,4] and this remains lacking for many drugs used in the perioperative period. The use of pharmacokinetic (PK) and pharmacodynamic (PD) mathematical equations (known as models) is a central foundation for improving dose estimation [25]. There are two covariates, size (reflecting body mass) and age (reflecting maturation processes in children [26]) that are important contributors accounting for PK parameter (e.g., clearance, volume) variability [27,28]. Contributors to pharmacodynamics (PD: E_MAX_, C_50_) variability are poorly quantified, but age, particularly in neonates, is important. Size can be standardized to a 70 kg person using allometric theory [29]. Fat is a component of body composition that certainly contributes to PK parameter variability [30], but has been poorly investigated in children [23].

We review the concepts behind the determination of dose for acetaminophen and endeavour to explain and quantify the impact of fat mass on dose computation.

## 4. Physiological Models

The pharmacokinetics of drugs used in children can be estimated using two quite different methods: physiological models (e.g., physiology-based pharmacokinetics, PBPK, top-down) or using patient data (e.g., compartment pharmacokinetic models, bottom-up). Physiological models use mathematical equations to describe an organism as a closed circulatory system consisting of compartments that represent the organs important for pharmacokinetic description, such as absorption, distribution, metabolism and elimination [31]. They describe the anatomy, physiological processes, and chemical reactions happening in the body e.g., blood flow to organs and their metabolic activity. These models have the capacity for incorporation of so much more information, such as pharmacogenomics or environmental influences [31,32]. Physiological models are capable of using existing information about obesity-related physiological changes (e.g., altered organ size, composition, and function), and drug-specific properties (e.g., lipophilicity and elimination pathways) [33]. This type of modelling has been used successfully to investigate clindamycin, trimethoprim/sulfamethoxazole, and metformin to better understand the dosing of these drugs in children with obesity [34,35].

This modelling technique has been used to investigate acetaminophen pharmacokinetics in humans [36] as well as special populations such as pregnant patients [37] and premature neonates [38]. It is lacking for obese children, but has potential use in the investigation of pharmacokinetics in these children. Enoxyparin dose, for example, has been investigated in obese children using both PBPK [39] and compartment models [40]; both methods concluded that fat-free mass was a good size scaler to use for Enoxyparin dose estimation in obese children.

## 5. The Target Effect

The determination of dose in children cannot be made using pharmacokinetic knowledge alone. It is necessary to understand pharmacodynamics as well. A key aspect of dose determination is knowing what concentration should be targeted in order to achieve the desired effect. This target concentration strategy [41] requires an understanding of the concentration–response relationship.

A concentration–analgesic response relationship has been described in children who have been given acetaminophen for pain after tonsillectomy. This relationship has been defined using a pharmacodynamic (PD) model, the E_MAX_ or Hill equation [42,43]:(1)$$Analgesic\;Effect=E_{MAX}\frac{C^{HILL}}{C_{50}^{HILL}+C^{HILL}}$$

The pharmacodynamic parameter, E_MAX_, is the maximum drug effect (5.17 on a visual analogue scale 0–10), C_50_ is the concentration eliciting half of E_MAX_ (9.97 mg/L), and the Hill exponent (Hill or N = 1) describes the steepness of the concentration–response curve [44]. This relationship is displayed graphically in Figure 1 and can be used to predict the target concentration known to be associated with a target effect. A target effect of 2.6 pain unit reduction (VAS 0–10) is associated with a target concentration of 10 mg/L. This acetaminophen target concentration of 10 mg/L is similar for both neonates and children [44,45]. Acetaminophen is a mild analgesic with a maximum effect of only 5.17 pain units. Children with an initial pain score of 10 pain units will still require remedication at this target concentration because their pain score will remain high at a VAS of 7.4 units. However, pain in those with an initial score of 6 pain units will be better managed.

## 6. Dose Calculation Using the Target Concentration

The target effect is the goal of drug treatment. This target effect is associated with a target concentration. The pharmacokinetic model can be used to calculate a dose [41,46] that achieves the target concentration [47], a process known as the target concentration strategy. Enteral acetaminophen disposition can usually be described using a single compartment [44,48,49,50]. Time–concentration relationships for a one-compartment pharmacokinetic model, for example, are expressed in terms of the parameter clearance (CL) and volume of distribution (V):(2)Concentration=doseV×e−time×CLV

The pharmacokinetic parameter volume of distribution (V) is used determine a loading dose that achieves a desired target concentration for a one-compartment model (Equation (3)) while clearance (CL) determines the maintenance dose rate (Equation (4)).
(3)Loading Dose=V×Target Concentration
(4)Maintenance Dose Rate=CL×Target Concentration

Acetaminophen administered enterally is often described using a one-compartment model, but that model may be inadequate to portray the time–concentration profile for acetaminophen if oral absorption is slow or if delivered intravenously, where further compartments are required to describe the time course of drug concentration in plasma [26,51]. This multi-compartment model is required to describe drugs administered intravenously into the central compartment (V1) that then redistributes to peripheral compartments (V2, V3, etc.; Figure 1). The loading dose may be too small if based on V1 or too big if based on the volume of distribution at steady state (Vss). Redistribution takes place during loading dose administration. The peripheral compartment may differ between lean and obese children due to drug lipid solubility, further complicating dose calculation.

The concentration used to describe the observed response can be that in the effect compartment (Ce) rather than the plasma (Cp) [52]. This additional compartment (the effect compartment) accounts for time delays between plasma concentration and observed response (Figure 1). The delay between plasma and effect compartments is described by an equilibration half-time (T1/2keo) and this is approximately 53 min for acetaminophen (Figure 2) [44]. This delay has clinical implications. The drug should be given before the anticipated pain insult, or the dose should be managed so that effect compartment concentration is above the target level in the post-anaesthesia recovery room (PACU). The use of a loading dose achieves both these aims. The high plasma concentrations overshoot the target concentration and the target concentration in the effect compartment is reached earlier than if a standard maintenance dose (MD) is used (Figure 2). In addition, the larger loading dose (LD) ensures that effect compartment concentrations are above the target level for a longer duration than the if a standard maintenance dose is used (e.g., LD 30 mg/kg vs. MD 15 mg/kg)

## 7. Dosing Concepts in the Child

The principles behind dose estimation involve an understanding of pharmacokinetic parameters, clearance and volume. Weight (reflecting size) and age (contributing maturation of physiological processes) are the major covariates contributing to parameter variability in children, [28] but fat mass is also important and contributes to both these parameters, even in lean individuals. Maturation of physiological and anatomical processes has a greater impact in neonates and infants.

### 7.1. The Association between Size and Dose

Drugs are commonly dosed per kilogram of total body weight in children. That dose often changes with age so that the dose (per kilogram) is higher for a 2-year-old child than for a 10-year-old child. This is because the pharmacokinetic parameters (e.g., CL, V) that determine dose are based on size. It is important to separate out the impact of size so that other covariate influences (e.g., age-related changes, organ dysfunction, or obesity-associated changes) can be assessed.

Drug dose calculations are commonly made assuming a linear relationship between TBW and dose (Equation (5)):(5)Dose=DoseSTD·TBWWTSTD
where a standard dose (Dose_STD_) is that for a person of standard weight (WT_STD_ e.g., 70 kg). This equation demonstrates dosing commonly known as dosing per kilogram. However, it is widely known that maintenance doses expressed as mg/kg, as in Equation (4), are too small in children when compared to adults; this linear approach is not a suitable general method for drug dosing in children [53]. The maintenance dose should be based on clearance (Equation (3)), but clearance has a nonlinear, not linear relationship with size.

### 7.2. Allometry

The use of allometry introduces the nonlinear relationship between size and clearance (Figure 3). Allometry is the relationship between the size of an organism and its physiology (functional aspects), morphology (structural aspects), and life history (temporal aspects). The relationship between physiological traits (e.g., metabolic processes such as clearance) and structural components (e.g., blood volume or volume of distribution), and time-related processes (heart rate, respiratory rate, and drug half-life) and size has been used to scale pharmacokinetic parameters.

The log of basal metabolic rate plotted against the log of body weight produces a straight line with a slope of ¾ across species, with size changes that are 18 orders of magnitude. Similar relationships for volumes (e.g., blood volume) have a slope of 4/4, while time-related functions (e.g., heart rate) have a slope of ¼. Fractal geometry is used to mathematically explain this allometric scaling law [54,55]. Total drug clearance may be expected to scale to weight with an exponent of ¾ (Equation (6)) [56], so that clearance in a child can be predicted from that in an adult person of standard weight, which is 70 kg (WT_STD_):(6)CLchild=CLadult×FMATURATION×TBWWTSTD¾

Clearance maturation occurs in the first year of life and a function describing this maturation is required (Figure 3) during that period. Maturation is usually described using another function (F_MATURATION_) that uses age as an independent variable.

Figure 3 shows how clearance is less than might be expected from total body weight with the linear per kilogram model. The difference between the linear total body weight prediction and the allometric weight prediction increases with total body weight. Other bodyweight scalers shown in Figure 3 (body surface area, fat-free mass) are also curvilinear in nature, a relationship that exists in both obese and lean individuals [20]. There is no change at any one weight where a size scaler should be changed from one to another. The rate of clearance increase slows as size increases; consequently, the dose, when expressed per kilogram of total body weight without allometry, is invariably excessive.

Total body weight, without allometry, is a poor size scaler. Ideal body weight (IBW), which also has a non-linear relationship to clearance (i.e., rate of clearance increase slows as size increases), is currently the only alternative body size scaler to total body weight [57] mentioned in the British National Formulary for Children [58]. However, the calculation of IBW is not facile and there are five published methods available for its calculation [59]. There is a poor understanding of when and how to calculate IBW among paediatricians [60]. If IBW should be the preferred size scaler for the obese child, then uncertainty exists about an obese size measure at which it should be implemented. Even the definition of obesity changes with age and body mass index [61]. Ideal body weight does not account for fat mass and is not the best scaler for all drugs in which fat mass has a varying impact on PK parameters [4]. A better scaler would consider the impact of fat mass on pharmacokinetic parameters and have applicability to all children, lean or obese.

### 7.3. Fat Mass

It is suggested that 75% of excess weight in obese children is fat mass, and the remainder is lean mass [23], but the very definition of obesity in children relies on variability above a mean weight for age, and excess weight is poorly quantified [61]. It is assumed that increases in fat mass alter the distribution of lipophilic drugs and increases in lean mass alter drug clearance, but there is a paucity of evidence supporting these assumptions for most drugs [23]. The contribution of fat mass to pathology is not acknowledged. Investigators have used an assortment of size scalers to empirically explain the contribution of fat mass for individual drugs [24] with scant evidence for the superiority of one metric over another.

#### 7.3.1. Lean Body Mass

It has been asserted that lean body mass (LBM) (commonly used interchangeably with lean body weight (LBW) and fat-free mass (FFM)) is the optimal size scalar for many drugs used in anaesthesia during the perioperative period [62,63,64,65,66]. The merits of using LBM have been reviewed, with the conclusion that LBM is a good predictor of drug dose for all drugs [67]. This extension to all drugs remains unproven [4,21]. Direct comparison with other size scalers has rarely been undertaken. Lean body mass does not consider fat mass, which is known to have an effect on pharmacokinetic parameters.

#### 7.3.2. Normal Fat Mass

Any size scaler must account for fat mass and must be applicable to children of all weights. There seems little value in using total body weight for children who are lean and then switching to an alternative size scaler in those children classified as obese. The idea of adding a fraction of fat mass to FFM has been used to estimate the mass that best describes structure and function based on allometric scaling theory [47]. This mass has been called normal fat mass (NFM) [68]. NFM is calculated from FFM and FAT mass (Equation (7)):FAT = TBW − FFM(7)

The fraction of FAT (Ffat) that contributes to the structural (V) or functional (CL) size is specific to each drug (Equation (8)).
(8)NFM=FFM+Ffat×FAT

If Ffat is estimated to be zero then NFM is FFM, while if Ffat is 1 then NFM is TBW. Normal fat mass, used in conjunction with allometry, lies between FFM and TBW when used with allometry (Figure 3) and is specific for each drug. NFM may also differ when used for clearance and when used for volume of distribution. NFM requires the determination of Ffat. While the impact of this parameter (Ffat) has been established for the renal elimination pathway over a broad range of ages from premature neonates to adults [69], it has only been determined for a handful of drugs [68,70,71], one of which is acetaminophen [72]. The parameter Ffat was estimated as 1 for volume (i.e., TBW) and 0.82 for clearance. Total body weight can then be used as the size metric for an acetaminophen loading dose.

A negative value for Ffat for clearance (Ffat_CL_) might suggest organ dysfunction. Obesity is associated with organ dysfunction in the morbidly obese. Dexmedetomidine was noted to have a negative value for Ffat_CL_ in morbidly obese adults [73]. Although we might anticipate that Ffat increases with lipid solubility when used for volume of distribution, this has not yet been demonstrated.

## 8. Application of NFM Principles for Acetaminophen Dosing in Children

Once the impact of fat mass on pharmacokinetic parameters (CL, V) has been evaluated, then those pharmacokinetic parameters can be used in all children, lean or obese. It is not necessary to change to a different size scaler simply because the patient fulfils criteria that determine a specific grade of obesity.

### 8.1. Loading Dose

An acetaminophen loading dose is commonly given by mouth (per os, po) preoperatively. The use of rectal formulations was favoured in the past because of fears related to the aspiration of gastric contents during anaesthesia consequent to having an increased volume of fluid in the stomach. However, fasting guidelines for children presenting for routine procedures have been relaxed [74,75,76] and acetaminophen elixir is cleared from the stomach quickly [77]. Rectal formulations, although effective, require a larger dose because of reduced bioavailability [78] and are associated with high plasma concentration variability and slow absorption [48,79].

The appropriate size matrix for a loading dose of acetaminophen is total body weight because Ffat_VOL_ = 1. Simulated plasma concentrations attained after a loading dose of acetaminophen of 30 mg/kg in a 6-year-old, 20 kg child (FFM 16.4 kg, BMI 15.12 kg/m^2^) are shown in Figure 2. Effect compartment concentrations of 10 mg/L are achieved at 25 min and decrease below this concentration at 3.5 h. The loading dose (30 mg/kg) is the same for obese and lean children. Clearance, however, determines the duration of time that concentrations remain above 10 mg/L. Clearance increases with weight when expressed per kilogram, and so the duration of concentration above 10 mg/L in a 6-year-old, 40 kg child (FFM 24.7 kg, BMI 30.25 kg/m^2^) is longer (Figure 2). However, while concentrations might be estimated to be below 10 mg/L at 4 h 25 min in the lean child, the impact of adding Ffat = 0.82 to the simulation is minimal. Simulation using TBW (Ffat = 1) with allometry reveals a time below 10 mg/L at 4 h 15 min, a small analgesic difference because the concentration–response curve is shallow at that concentration (Figure 1). The impact of separating out fat mass for loading dose estimation is minimal. Total body weight is the better scaler for an acetaminophen loading dose.

### 8.2. Maintenance/Infusion Dose

The difference in drug clearance between an adult and a child is predictable from NFM used in conjunction with allometry (Equation (9)) [56]:(9)CLCHILD=CLADULT×NFMCHILDNFMADULT34

The maintenance dose can then be calculated based on the steady-state target concentration (Equation (10)):(10)Maintenance Dose=Clearance×Target Steady State Concentration

Simulation has been used to demonstrate the impact of size on predicted concentration in 10-year-old children (weight 30 kg, FFM 24 kg, BMI 15.3 kg/m^2^; weight 50 kg, FFM 33 kg, BMI 25.5 kg/m^2^; and weight 70 kg, FFM 39 kg, BMI 35.7 kg/m^2^ (Figure 4)) given a loading dose of 30 mg/kg and a maintenance dose of 15 mg/kg 6-hourly. While clearance increase is nonlinear and predicted concentrations increase with size, these higher concentrations are unlikely to contribute a meaningful improvement to analgesia. Similarly, the use of NFM (Ffat_CL_ = 0.82) instead of TBW (Ffat_CL_ = 1.0) will have minimal impact on pain scores. This clinical impact is minimal because predicted concentrations are on the flat part of the concentration–response curve, because a clinically important pain score change is more than 1 pain unit (VAS 0–10) [80,81], and because both PK and PD parameter estimates are associated with considerable variability [28].

If it is assumed that the target concentration in children and adults is the same, then the relationship between doses in children and adults can be predicted (Equation (11)):(11)Maintenance DoseCHILD=Maintenance DoseADULT×NFMCHILDNFMADULT34

This dosing extrapolation becomes problematic in teenagers because the adult maintenance dose is commonly capped at 1000 mg. An obese teenager (e.g., weight 125 kg) administered a loading dose of acetaminophen 2 g with a maintenance dosing of 1000 mg 6-hourly will not reach the target concentration of 10 mg/L at steady-state conditions (Figure 5). There will be a mean pain score decrease of 2 (VAS 0–10) and while this is a meaningful pain decrease, it is a small decrease and will require supplementation from other analgesic drugs.

## 9. Consideration of Adverse Effects

### 9.1. Hepatotoxicity

Acetaminophen (APAP) dosing in children is tempered by concerns of hepatotoxicity. The toxic metabolite of acetaminophen, N-acetyl-p-benzoquinone imine (NAPQI), is formed by the P450 hepatic cytochrome, CYP2E1. Hepatotoxicity is dependent on the balance between the rate of NAPQI formation, the capacity of the acetaminophen elimination clearance pathways involving hepatic glucuronide and sulfate conjugation, and the initial content and maximal rate of synthesis of hepatic glutathione that mops up NAPQI. NAPQI binds to intracellular hepatic macromolecules to produce cell necrosis and damage.

#### 9.1.1. Loading Dose

Hepatotoxicity in children is relatable to concentration, not dose. Dose, a measure determined using pharmacokinetic parameters, is commonly used as a surrogate to assess the risk of hepatotoxicity. There remains a distinction between hepatotoxicity due to a single dose of acetaminophen (e.g., a loading dose) and that administered over a duration longer than 2–3 days at doses greater than 90 mg/kg/min. The Rumack and Matthew [82] acetaminophen toxicity nomogram is widely used to guide the management of acetaminophen overdose in adults and children. This nomogram interprets acetaminophen clearance and relates clearance to a concentration at time points after 4 h. Acetaminophen concentrations of more than 300 mg/L at 4 h were always associated with severe hepatic lesions, but none were observed in adults with concentrations lower than 150 mg/L. The half-life was less than 4 h in all patients without liver damage.

Clearance, expressed as L/h/kg, is greater in children than in adults. The 4 h concentration determined by clearance in preschool children following accidental ingestion of acetaminophen elixir is less than that in adults. This is because the absorption of the elixir is more rapid than that noted in adults following tablet ingestion and because clearance is faster in toddlers than in adults. As a consequence, younger children (1–5 years) require larger doses than older children and adults to achieve similar concentrations at 4 h. Children (1–5 years) with reported accidental ingestion of greater than 250 mg/kg (compared to 150 mg/kg in adults) can have the serum concentration measured at 2 h after ingestion rather than the 4 h time point recommended in adults [83]. Preoperative loading doses administered before anaesthesia induction (e.g., 30 mg/kg) are well below those associated with toxicity.

#### 9.1.2. Maintenance Dose

Maintenance dosing in excess of 90 mg/kg/day administered over 2–3 days is of greater concern. This dose will cause higher concentrations in teenagers than in children 2–3 years of age because clearance, expressed per kilogram, is faster in the younger cohort. Hepatic and renal disease, malnutrition, and dehydration may increase the propensity for toxicity. Medications that induce NAPQI formation (e.g., phenobarbitone, phenytoin, and rifampicin) may also increase the risk of hepatotoxicity. The influence of obesity on acetaminophen toxicity is unknown and although there are occasional reports [84], the influence of dose administered per kilogram and the underlying pathology of the child remains uncertain. Hepatotoxicity causing death or requiring liver transplantation has been reported with doses above 75 mg/kg/day in children and 90 mg/kg/day in infants. It has been suggested that even these traditional regimens may cause hepatotoxicity if used for longer than 2 to 3 days [85]. The dose that might lead to hepatotoxicity remains speculative and ‘safe’ doses range from 60 mg/kg/day through to 90 mg/kg/day [86,87].

### 9.2. Concentration or Dose

Dosing restrictions for acetaminophen have eventuated because of toxicity fears associated with doses greater than 75 mg/kg/day in children. This lower dose became standard clinical practice following the introduction of the intravenous formulation of acetaminophen, where the dose was dictated by the pharmaceutical industry operating in a litigious environment. It is not the dose that causes toxicity, but rather a plasma concentration (or perhaps exposure, measured using the area under the curve). The dose is determined using pharmacokinetic knowledge to enable a target concentration to be reached. Fear of litigation has resulted in underdosing of obese teenagers.

### 9.3. The Acetaminophen–NSAID Interaction

Practitioner choices for acetaminophen maintenance dosing are limited. If the use of the target concentration is chosen, then a lower target concentration than 10 mg/L must be used, resulting in less effective analgesia. Capping a dose at 1000 mg in obese teenagers is particularly irksome for practitioners when it is known that a target concentration of 10 mg/L cannot be achieved with the 1000 mg dose. This concentration of 10 mg/L is not associated with toxicity but has a reasonable analgesic effect.

One solution is to use acetaminophen–nonsteroidal anti-inflammatory drug (NSAID) combination therapy. Acetaminophen and NSAIDs are often given together for the management of pain [88] or fever [89]. They can be safely combined using lower doses of each drug without increases in their associated adverse effect profiles and combination therapy is both popular [90] and recommended for analgesia after procedures such as tonsillectomy [91]. The maximum analgesic effect (e.g., E_MAX_ 5 to 6, VAS 0–10) remained the same as that for either agent alone, but that analgesic effect was sustained at 4 to 8 h after combination dosing [92,93,94]. The dose for this combination therapy is often also dictated by regulatory authorities, e.g., 4.5 mg/kg ibuprofen (maximum dose 300 mg) and 15 mg/kg paracetamol (max dose 1000 mg) [95]. The dose of ibuprofen is lower than that commonly prescribed alone [96]. Ibuprofen has a similar FAT fraction for clearance (fFat_CL_ = 0.86) to acetaminophen, but with a lower FAT fraction for volume (fFat_VOL_ = 0.72) [72]. However, because the ibuprofen dose is small, the dose for the mixture could be calculated based on calculations for acetaminophen alone.

There are few data concerning ibuprofen pharmacokinetics or dosing in obese children [97]. Ibuprofen dose could be based on normal fat mass with allometry. Should ibuprofen be given separately from acetaminophen, then the dose of ibuprofen can be readily calculated using NFM (Equation (9)). Fat-free mass (FFM) can be predicted from sex, height, and total body weight (Equation (12)).
(12)FFM=WHSMAX×HT2×TBWWHS50×HT2+TBW
where WHS_MAX_ is the maximum FFM for any given height (HT, m) and WHS_50_ is the TBW value when FFM is half of WHS_MAX_. For men, WHS_MAX_ is 42.92 kg·m^−2^ and WHS_50_ is 30.93 kg·m^−2^, and for women, WHS_MAX_ is 37.99 kg·m^−2^ and WHS_50_ is 35.98 kg·m^−2^ [98]. Computation of FFM in children has been simplified by the availability of online calculators (e.g., [99]).

## 10. Conclusions

The loading dose is determined by total body weight (mg/kg) and is the same in both lean and obese children. The maintenance dose (mg/kg) in children with obesity is less than that presumed using linear scaling. There is a curvilinear relationship between clearance and weight. The dose should reflect that relationship. Ideal body weight (IBW) has been proposed as an appropriate size scaler for use in obese children. However, IBW calculation is not easy, and although it may describe a curvilinear relationship with clearance, it is neither drug-specific nor does it distinguish between clearance or volume. The fat mass has an influence on both clearance and volume and fat mass is present in children, even those considered lean.

The use of NFM as a size scaler for acetaminophen has merit, but the computations required to calculate the dose are also not facile, although online calculators are available. Acetaminophen has the advantage that the loading dose can be calculated using TBW. The maintenance dose can be better calculated using NFM; however, because the FAT fraction is large (Ffat_CL_ = 0.82) and because some consider the estimation of this parameter to have low precision [100], then TBW is a reasonable proxy if used with allometric scaling.

If we assume a typical adult (70 kg) will be given a maintenance dose of 1000 mg (15 mg/kg) four times a day, then the dose for a 10-year-old weighing 30 kg can be readily calculated (Equation (14)):(13)DOSEchild=DOSEadult(1000 mg)×30 kg70 kg¾~500 mg

A 10-year-old child 120 kg in weight will require a maintenance dose (Equation (13))
(14)DOSEchild=DOSEadult(1000 mg)×120 kg70 kg¾~1500 mg

It can be noted that while weight has increased 4-fold from 30 kg to 120 kg, the dose has only increased 3-fold from 500 mg to 1500 mg, exemplifying the non-linear relationship between clearance and weight. The dose can be scaled directly from the adult dose using allometry in children, but not for infants and neonates where physiological processes are maturing.

## Figures and Tables

**Figure 1 children-10-00625-f001:**
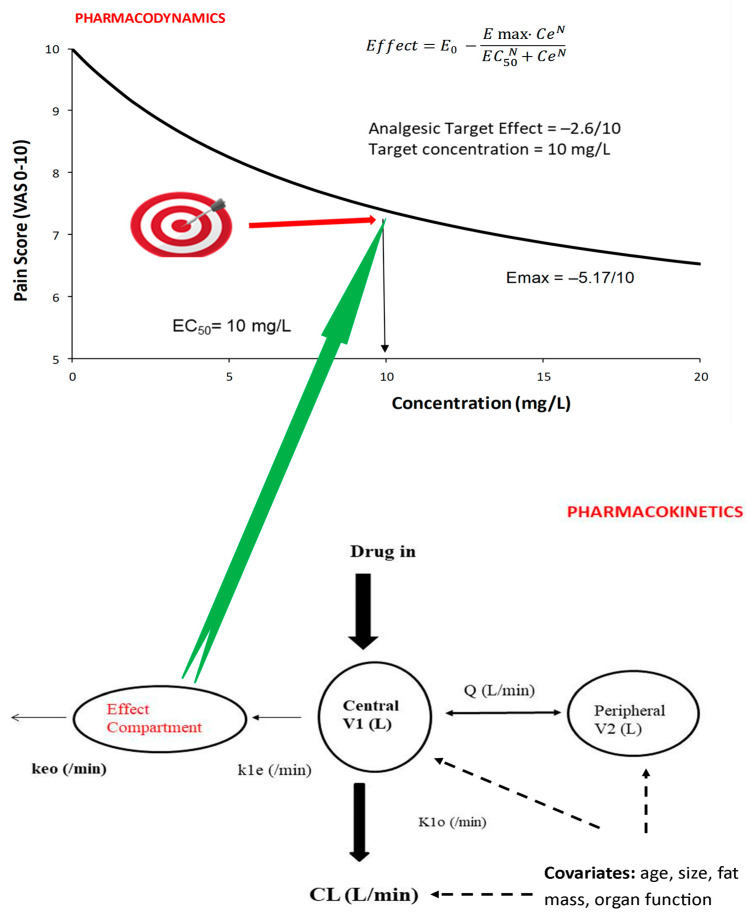
A diaphragmatic representation of the target concentration strategy. The upper panel shows the concentration–response for acetaminophen and analgesia. This response is described mathematically using the E_MAX_ equation. The target effect of 2.6 pain unit reduction (VAS 0–10) is associated with a target concentration of 10 mg/L. A 2-compartment pharmacokinetic model (lower panel) is used to calculate a dose that achieves this target concentration in the effect compartment (Ce). Concentration in the central compartment (Cp) is linked to that in the effect compartment by a rate constant (k1e = keo at steady-state conditions). This equilibration rate constant (keo, determining rate from effect compartment to outside) is often expressed as the equilibration half-time (T_1/2_keo).

**Figure 2 children-10-00625-f002:**
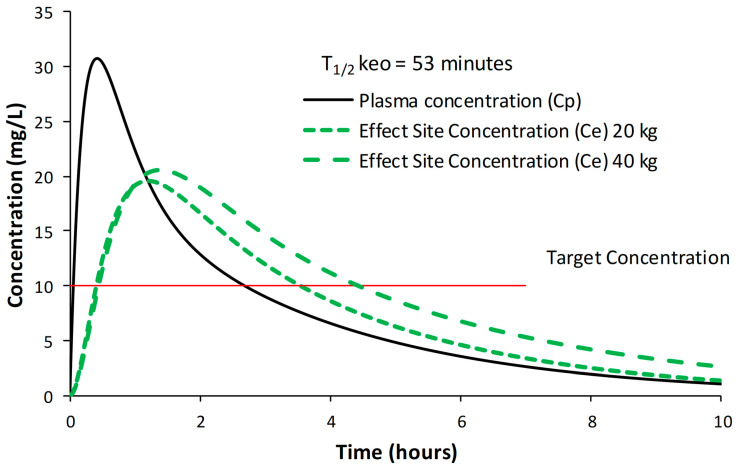
Simulated time–concentration profiles are shown for a loading dose of acetaminophen of 30 mg/kg in a 6-year-old, 20 kg child (FFM 16.4 kg, BMI 15.12 kg/m^2^). Effect compartment concentrations of 10 mg/L are achieved at 25 min and decrease below this concentration at 3.5 h. While the loading dose (30 mg/kg) is the same for obese and lean children, clearance determines the duration of time that concentrations are above 10 mg/L. The duration of concentration above 10 mg/L in a 6-year-old, 40 kg child (FFM 24.7 kg, BMI 30.25 kg/m^2^) is longer.

**Figure 3 children-10-00625-f003:**
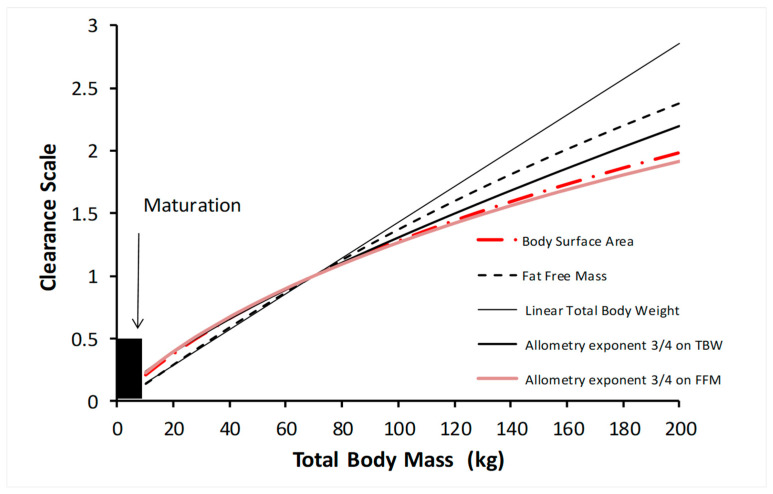
Changes in clearance are demonstrated as total body mass (expressed as weight) increases. The size metrics (body surface area, fat-free mass, linear total body weight, total body weight with allometry, and fat-free mass with allometry) are shown relative to a person with 70 kg total body mass. Children younger than 1 year of age (approx. 10 kg) are not shown because maturation is incomplete in that cohort. There is a nonlinear relationship between weight and clearance for most body size metrics, demonstrated with a curvilinear shape. The per kilogram model is shown as a straight line and increasingly overestimates clearance in adults of weight greater than 70 kg.

**Figure 4 children-10-00625-f004:**
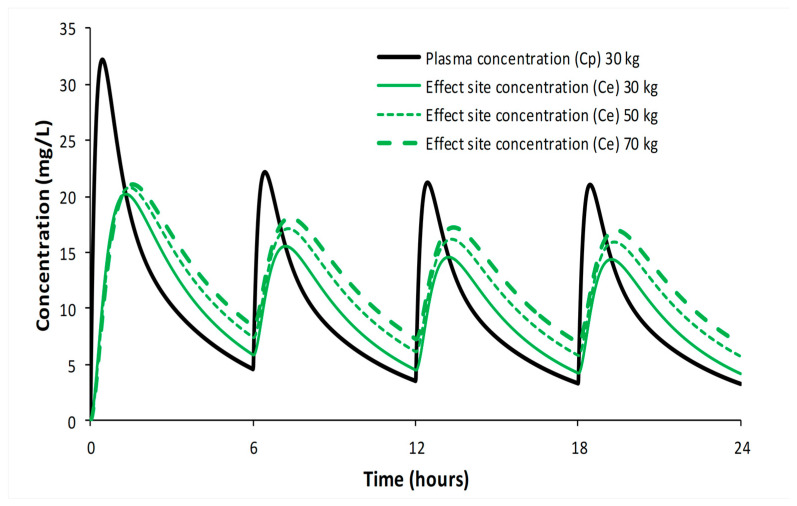
Simulated time–concentration profiles are shown for a loading dose of acetaminophen of 30 mg/kg in a 10-year-old, 30 kg child (FFM 24 kg, BMI 15.3 kg/m^2^). Effect compartment concentrations for that child and obese children (weight 50 kg, FFM 33 kg, BMI 25.5 kg/m^2^ and weight 70 kg, FFM 39 kg, BMI 35.7 kg/m^2^) are shown after regular maintenance dosing of 15 mg/kg 6-hourly. Concentration increases as weight increases because clearance has a nonlinear relationship with weight.

**Figure 5 children-10-00625-f005:**
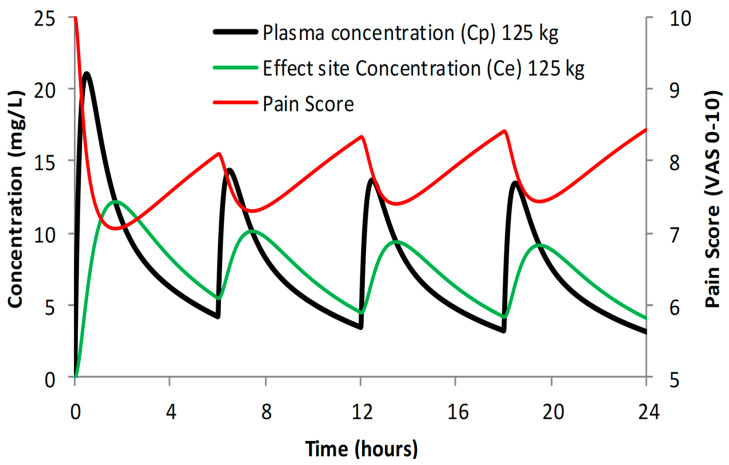
Simulation to demonstrate that an obese teenager (125 kg) administered a loading dose of acetaminophen of 2000 mg with a maintenance dosing of 1000 mg 6-hourly will not reach the target concentration of 10 mg/L at steady-state conditions. There will be a mean pain score decrease of 2 (VAS 0–10) and while this is a meaningful pain decrease, it is a small decrease and will require supplementation from other analgesic drugs.

## Data Availability

No new data were generated or analysed in this study. Data sharing is not applicable to this article.

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
