# Peer review of "Perioperative Acetaminophen Dosing in Obese Children"

_children, 2023, doi:10.3390/children10040625_

Round 1

Reviewer 1 Report

The article addresses an interesting topic for the field of pediatrics, that of calculating and adapting the dose of acetaminophen for children in the perioperative period. Adapting the dose according to body weight, but also according to body fat is a correct approach to calculating the dose useful in practice. Fat the body has an influence on clearance as well as on liver function.

QUESTIONS:

1. Why was the analysis done only for the use of acetaminophen in the perioperative period

2. Can the results of the analyzes be applied to other pathologies that require acetaminophen?

The article addresses an interesting topic for the field of pediatrics, that of calculating and adapting the dose of acetaminophen for children in the perioperative period. Adapting the dose according to body weight, but also according to body fat is a correct approach to calculating the dose useful in practice. Fat the body has an influence on clearance as well as on liver function.

QUESTIONS:

1. Why was the analysis done only for the use of acetaminophen in the perioperative period

2. Can the results of the analyzes be applied to other pathologies that require acetaminophen?

The article addresses an interesting topic for the field of pediatrics, that of calculating and adapting the dose of acetaminophen for children in the perioperative period. Adapting the dose according to body weight, but also according to body fat is a correct approach to calculating the dose useful in practice. Fat the body has an influence on clearance as well as on liver function.

QUESTIONS:

1. Why was the analysis done only for the use of acetaminophen in the perioperative period

2. Can the results of the analyzes be applied to other pathologies that require acetaminophen?

The article addresses an interesting topic for the field of pediatrics, that of calculating and adapting the dose of acetaminophen for children in the perioperative period. Adapting the dose according to body weight, but also according to body fat is a correct approach to calculating the dose useful in practice. Fat the body has an influence on clearance as well as on liver function.

QUESTIONS:

1. Why was the analysis done only for the use of acetaminophen in the perioperative period

2. Can the results of the analyzes be applied to other pathologies that require acetaminophen?

The article addresses an interesting topic for the field of pediatrics, that of calculating and adapting the dose of acetaminophen for children in the perioperative period. Adapting the dose according to body weight, but also according to body fat is a correct approach to calculating the dose useful in practice. Fat the body has an influence on clearance as well as on liver function.

QUESTIONS:

1. Why was the analysis done only for the use of acetaminophen in the perioperative period

2. Can the results of the analyzes be applied to other pathologies that require acetaminophen?

The article addresses an interesting topic for the field of pediatrics, that of calculating and adapting the dose of acetaminophen for children in the perioperative period. Adapting the dose according to body weight, but also according to body fat is a correct approach to calculating the dose useful in practice. Fat the body has an influence on clearance as well as on liver function.

QUESTIONS:

1. Why was the analysis done only for the use of acetaminophen in the perioperative period

2. Can the results of the analyzes be applied to other pathologies that require acetaminophen?

The article addresses an interesting topic for the field of pediatrics, that of calculating and adapting the dose of acetaminophen for children in the perioperative period. Adapting the dose according to body weight, but also according to body fat is a correct approach to calculating the dose useful in practice. Fat the body has an influence on clearance as well as on liver function.

QUESTIONS:

1. Why was the analysis done only for the use of acetaminophen in the perioperative period

2. Can the results of the analyzes be applied to other pathologies that require acetaminophen?

The article addresses an interesting topic for the field of pediatrics, that of calculating and adapting the dose of acetaminophen for children in the perioperative period. Adapting the dose according to body weight, but also according to body fat is a correct approach to calculating the dose useful in practice. Fat the body has an influence on clearance as well as on liver function.

QUESTIONS:

1. Why was the analysis done only for the use of acetaminophen in the perioperative period

2. Can the results of the analyzes be applied to other pathologies that require acetaminophen?

Author Response

Response to reviewer 3

The article addresses an interesting topic for the field of pediatrics, that of calculating and adapting the dose of acetaminophen for children in the perioperative period. Adapting the dose according to body weight, but also according to body fat is a correct approach to calculating the dose useful in practice. Fat the body has an influence on clearance as well as on liver function.
QUESTIONS:
1.    Why was the analysis done only for the use of acetaminophen in the perioperative period
Acetaminophen is a nice practical example. 
a)    It is the commonest paediatric analgesic in the world
b)    Toxicity fears place limitations on dose. It is well recognised that obese teenagers are underdosed
c)    There is misunderstanding about the major contributing factor to toxicioty and that is concentration. Dose is simply a surrogate for toxicity. It is not a valid surrogate in the obese because increased dose does not correlate with increased concentration
d)    Acetaminophen has a Ffat that approaches 1, so TBW is actually not a bad substitute for NFM. 
e)    NFM is hard to compute , but use of TBW easily demonstrates the principles behind dose and the “rule of diminishing returns”. Doubling of weight does not mean doubling of dose. That, I suspect is the majot miscommunication
I had originally intended to include NSAIDs in this review. However, the decision was made to simply constrain the paper to acetaminophen, in order extound the principles behind dosing in the obese. I have mentioned the Ffat estimates for ibuprofen, but have not gone into dosing. However, I do briefly mention dosing for acetaminophen/ibuprofen combination therapy. Dose of ibuprofen is actually low (5 mg/kg) so that use of acetaminophen component of the mixture to determine dose is fine.
I have very briefly expanded on use of NFM to determine ibuprofen dose in obese children. However, that is probably as complex as use of IBW. It requires formulae for FFM in children (male and female) and that is now included. 
2.    Can the results of the analyzes be applied to other pathologies that require acetaminophen?
The answer is yes and no! This current analysis reviews covariates (age, size and fat mass) and their impact on clearance and volume. This serves as a baseline. Significant other pathology (eg hepatic failure) would be a further covariate acting on this “baseline”

Reviewer 2 Report

Thank you for your submission to Children. This review is a comprehensive an in-depth description
of acetaminophen dosing in children, focussing on the obese child. The authors are very
knowledgeable and have very eloquently discussed the relevant concepts. Some concepts would be
better discussed in a practical sense that can be applicable to the bedside clinician. Some suggested
changes are detailed below.
General
Please be clear when using the word volume and volume of distribution -both have been used
seemingly interchangeably in the manuscript; rather state volume of distribution using the accepted
abbreviation Vd.
Size and weight are also used interchangeably in the discussion (L57 weight scalar vs L63 size scalar).
The several places in the manuscript where this is done. Please use one descriptor throughout.
How should the reader define the obese child? Using BMI (for which age groups) or growth centiles?
Abstract
L12: please state the suggested loading dose
L13: please state the maintenance dose
These two doses should be discussed in the text as a starting point before explaining the implications
in the obese child.
L28: target concentration. Consequently, ...
L29-30: Acetaminophen is commonly used for longer than 3 days as an adjunct analgesic. Are you
suggesting that this should not be done? This would not be a practical suggestion considering that it
forms part of the basic analgesic options for most patients and it is cheap and readily accessible,
especially in a resource limited environment.
Body
L57: smorgasbord – change word
L131: lean children weight results
L144: larger loading dose
L145: than if a standard dose
L247: classified
Consideration for adverse effects: The maintenance dose of 75mg/kg can be easily reached using a
LD of 30mg/kg and four MDs of 10mg/kg. What recommendation do you make regarding the MD in
the obese child? In the next paragraph the debate for concentration or dose restriction is made.
What concentration does your proposed dose correlate to?
L413: irksome – change word
L417: What dosing scalar will you suggest for acetaminophen in the combined formulation with
NSAIDs?

Author Response

Reviewer 1

Thank you for your submission to Children. This review is a comprehensive an in-depth description
of acetaminophen dosing in children, focussing on the obese child. The authors are very
knowledgeable and have very eloquently discussed the relevant concepts. Some concepts would be
better discussed in a practical sense that can be applicable to the bedside clinician. Some suggested
changes are detailed below.

General
Please be clear when using the word volume and volume of distribution -both have been used
seemingly interchangeably in the manuscript; rather state volume of distribution using the accepted
abbreviation Vd.

**This nomenclature always causes a spot of bother. Volume of distribution (Vd) is correct terminology, but is usually applied to a one compartment model or to PK summary statistics determined using non-compartment models. The term fails when describing multicompartment models (e.g. V1, V2 rather than Vd) or even volume at steady state (Vss rather than Vd); all of which are volumes of distribution. Instead, the word volume is simply used to describe the many different volumes that bemuses and confuses readers. We have used the abbreviation V for volume of distribution as a general descriptive term, rather than Vd that usually applies to a one-compartment model. . This avoids specifying which volume of distribution. Allometric scaling is used on all volumes. Similarly we use the symbol CL for all clearances (total body clearance and intercompartment clearances, Q1, Q2)

Size and weight are also used interchangeably in the discussion (L57 weight scalar vs L63 size scalar).
The several places in the manuscript where this is done. Please use one descriptor throughout.

**The reviewer is correct. Size scalers are not all based on weight.

**The other tricky thing is the use of mass or weight to describe these scalers and the two measures are used interchangeable in the literature. Weight is the better known measure. We have elected to use TBW for routine weight but TBM when used with allometry. We have elected to use whichever size scaler is in common use e.g. IBW but FFM

How should the reader define the obese child? Using BMI (for which age groups) or growth centiles?

**Assessment of obesity is not easy because body composition changes with age, as does the fat mass. The definition of obesity in children is based on body mass index (BMI) changes with age (Growth Charts - Clinical Growth Charts (cdc.gov)). The BMI index is less in childhood than infancy. There is a nidir at 6-7 years and it increases with subsequent age. While adult obesity might be classified a grade 1 (BMI>30 kg/m2), grade 2 (BMI>35 kg/m2) and grade 3 (BMI>40 kg/m2) child obesity is graded depending on the percentile above the median for that particular age group; overweight (>85th percentile), obese (>95th percentile) and severely obese 120% of 95th percentile or >35 kg/m2).

**This is further complicated by an increasing median value in some first world countries! The advantage of the allometric model proposed in this paper is that dose is not determined by an obese scale. The model can be used for both lean and obese. There is no sudden shift from TBW to IBW at an arbitrary size. Similarly, FFM is calculated by knowing weight and height alone. FFM applies to all children independent of size status.

**We now specifically mention the difficulty assessing obesity gradation and how that changes with age. This is important because some scalers are advocated only in the obese, while simple perkilo models are used in the lean.

Abstract
L12: please state the suggested loading dose

**We are discussing principles for acetaminophen dosing.in this paper. The loading dose used or not used will be dependent on type of surgery and local input from clinicians, pharmacists, anesthetists etc. We discuss, with simulation, the loading dose typically used in our own departments, but we do not wish to dictate what others should do. We outline that the loading dose should achieve a target concentration commensurate with a degree of analgesia in the recovery room. This will depend on duration of surgery, time dose administered, dose amount etc
L13: please state the maintenance dose

**Local pharmaceutical policy dictates maintenance dose. We in NZ are quite happy with 90 mg/kg/day. Europe and USA have far stricter policies around dose; dictated by EMA or FDA. This aspect is discussed later in relation to the target concentration. If the target concentration cannot be achieved because of dose restriction, then combination therapy (with NSAIDs) may be used. Alternatively a lower degree of analgesia accepted.

These two doses should be discussed in the text as a starting point before explaining the implications
in the obese child.

**We disagree. Dose is of secondary consideration after PKPD principles that can be used to determine dose. We are introducing readers to the principles behind dose determination in the obese child.

L28: target concentration. Consequently, ...

**Comma added
L29-30: Acetaminophen is commonly used for longer than 3 days as an adjunct analgesic. Are you
suggesting that this should not be done? This would not be a practical suggestion considering that it
forms part of the basic analgesic options for most patients and it is cheap and readily accessible,
especially in a resource limited environment.

**Acetaminophen is used for months in some children (e.g., rheumatoid arthritis). It is the backbone of pediatric analgesia, both acute and chronic. However, hepatotoxicity is reported after 2-3 days using doses as low as 75 mg/kg/day. Consequently, some regulatory authorities restrict dose down to 75 mg/kg/day or even 60 mg/kg/day. This trend to lower doses has been exacerbated since the IV formulation was introduced where dose was limited to 60 mg/kg/day.  

Body
L57: smorgasbord – change word

**We thought smorgasbord a good choice of word. The size scalers are arrayed in front of the practitioner like a breakfast selection. We could of course use the word salmagundi or even potpourri. Alas, we have elected to use “assortment”

L131: lean children weight results

**This was a confusing sentence. We have reworded… The peripheral compartment may differ between lean and obese children due to drug lipid solubility, further complicating dose calculation.

L144: larger loading dose

**The loading dose is larger than the usual maintenance dose
L145: than if a standard dose

**changed to maintenance dose
L247: classified

**Spelling corrected for classified
Consideration for adverse effects: The maintenance dose of 75 mg/kg can be easily reached using a
LD of 30mg/kg and four MDs of 10mg/kg.

**It is concentration that causes effect and adverse effect. We make this point repeatedly in the paper. Dose is a surrogate index for concentration. The Loading dose simply achieves the target concentration. The daily maintenance dose, that maintains the concentration, is separate from LD. Maintenance dosing only causes problems when doses are maintained for 3-4 days at “higher doses”.. The loading dose is immaterial to that toxicity

 and what  recommendation do you make regarding the MD in the obese child?

**Use allometry to calculate the dose based on local “rules”. If the maintenance dose is 1000 mg (15*70) 4 times a day in an adult (70 kg) then maintenance dose is

**childdose=1000 mg x (WT/70)^0.75

**We have included examples (a 10-year old child weighing 30 kg and another weighing 120 kg using this equation in the concluding paragraph

In the next paragraph the debate for concentration or dose restriction is made.
What concentration does your proposed dose correlate to?

**Concentrations for a 10 year old child of weight 30 kg, 50 kg and 70 kg are shown in Figure 4. These children were given a loading dose 30 mg/kg and a maintenance dose 15 mg/kg 6 hourly

L413: irksome – change word

**It is irksome for practitioners to cap a dose when they know therapeutic concentrations will not be achieved. When a larger dose is prescribed practitioners are abused by pharmacy staff, nursing staff and even parents. It is irksome. Synonyms include annoying, bothersome, irritating, exasperating, tiresome, trying, tedious, vexing. The sentence has been altered …”particularly irksome for practitioners when it is known that..”

L417: What dosing scalar will you suggest for acetaminophen in the combined formulation with
NSAIDs?

**Formulation for acetaminophen-ibuprofen exist. Both the proportions of drug in the mixture and the maximum dose allowed are proscribed by regulatory authorities. For example the EMA suggests ibuprofen 4.5 mg/kg (maximum dose 300 mg) and paracetamol 15 mg/kg (max dose 1000 mg). Analgesic response is dependent on a concentration-response relationship but this response is plotted on a surface response curve. Different combinations may achieve the same result; bigger doses achieve the same Emax, but duration of effect is longer.

**Ibuprofen has an fFatCL=0.86 with an fFatVOL = 0.721. However prescribed dose is small2 and the dose of the mixture could be scaled using acetaminophen dose from within the mixture. This aspect is now discussed in text.

  1. Morse JD, Stanescu I, Atkinson HC, Anderson BJ. Population Pharmacokinetic Modelling of Acetaminophen and Ibuprofen: the Influence of Body Composition, Formulation and Feeding in Healthy Adult Volunteers. Eur J Drug Metab Pharmacokinet 2022;47:497-507.
  2. Anderson BJ, Hannam JA. A target concentration strategy to determine ibuprofen dosing in children. Pediatr Anesth 2019;29.

Reviewer 3 Report

I'd like to thanks the authors for their grate work and effort in writing the review article manuscript entitled "Perioperative Acetaminophen Dosing In Obese Children". The manuscript reviewed acetaminophen which is very commonly used drug worldwide specially about the children and obese children. The manuscript is well written in clear language. However, I have some minor points that might improve the current version. 

- In figure 1 (lower panel) there was a left arrow after "effect compartment". Could you clarify where this arrow goes? or you might delete it if it is unnecessary. 

- some equations (e.g., 5, 9, 10, and 11) has table borders. Is there any specification for that? or just make the borders with while/no color.

- Also Figure 3 has a gray border.

- In line 211, when you refer to (Figure 3, FAGE). Adding FAGE was not clear to me. You might delete FAGE and refer to the age in the next sentence. 

Author Response

Reviewer 2

I'd like to thanks the authors for their great work and effort in writing the review article manuscript entitled "Perioperative Acetaminophen Dosing In Obese Children". The manuscript reviewed acetaminophen which is very commonly used drug worldwide specially about the children and obese children. The manuscript is well written in clear language. However, I have some minor points that might improve the current version. 

- In figure 1 (lower panel) there was a left arrow after "effect compartment". Could you clarify where this arrow goes? or you might delete it if it is unnecessary. 

**This is the rate constant keo for movement from the effect compartment to the outside. It is the same as k1e at steady state. It is commonly expressed as T1/2keo, the equilibration half-time. The arrow is currently labelled in the figure. We have expanded the explanation in the legend. I should say that this rate constant causes endless confusion with students and even colleagues.

- some equations (e.g., 5, 9, 10, and 11) has table borders. Is there any specification for that? or just make the borders with while/no color.

**Corrected

- Also Figure 3 has a gray border.

**I am hoping that type setters can correct this, if troublesome. It could easily be cropped, for example.

- In line 211, when you refer to (Figure 3, FAGE). Adding FAGE was not clear to me. You might delete FAGE and refer to the age in the next sentence. 

**The reviewer is correct. FAGE changed to FMATURATION. Maturation is usually described using another function (FMATURATION) that uses age as an independent variable
